# Novel Roles of MT1-MMP and MMP-2: Beyond the Extracellular Milieu

**DOI:** 10.3390/ijms23179513

**Published:** 2022-08-23

**Authors:** Deanna V. Maybee, Nicole L. Ink, Mohammad A. M. Ali

**Affiliations:** Department of Pharmaceutical Sciences, SUNY Binghamton University School of Pharmacy and Pharmaceutical Sciences, Binghamton, NY 13902-6000, USA

**Keywords:** matrix metalloproteinases (MMPs), MMP-2, MT1-MMP, intracellular roles, subcellular localization

## Abstract

Matrix metalloproteinases (MMPs) are critical enzymes involved in a variety of cellular processes. MMPs are well known for their ability to degrade the extracellular matrix (ECM) and their extracellular role in cell migration. Recently, more research has been conducted on investigating novel subcellular localizations of MMPs and their intracellular roles at their respective locations. In this review article, we focus on the subcellular localization and novel intracellular roles of two closely related MMPs: membrane-type-1 matrix metalloproteinase (MT1-MMP) and matrix metalloproteinase-2 (MMP-2). Although MT1-MMP is commonly known to localize on the cell surface, the protease also localizes to the cytoplasm, caveolae, Golgi, cytoskeleton, centrosome, and nucleus. At these subcellular locations, MT1-MMP functions in cell migration, macrophage metabolism, invadopodia development, spindle formation and gene expression, respectively. Similar to MT1-MMP, MMP-2 localizes to the caveolae, mitochondria, cytoskeleton, nucleus and nucleolus and functions in calcium regulation, contractile dysfunction, gene expression and ribosomal RNA transcription. Our particular interest lies in the roles MMP-2 and MT1-MMP serve within the nucleus, as they may provide critical insights into cancer epigenetics and tumor migration and invasion. We suggest that targeting nuclear MT1-MMP or MMP-2 to reduce or halt cell proliferation and migration may lead to the development of new therapies for cancer and other diseases.

## 1. Introduction

Matrix metalloproteinases (MMPs) are a family of endopeptidases that consists of 28 enzymes in vertebrates, 24 genes of which are found in humans [1]. These enzymes are distinguishable by their zinc-dependent activation sites in addition to their known tendency to proteolyze extracellular matrix (ECM) components [2,3]. The majority of MMPs have similar domains homologous to MMP-1, the initial MMP discovered in the early 1960s, and are inhibited by the tissue inhibitor of MMPs (TIMPs), with a few other identifiable characteristics [4,5]. Structurally, these proteins consist of a pro-peptide domain that is removed extracellularly for activation, a zinc-dependent catalytic domain, and, in most MMPs, a hemopexin-like C-terminal that is important for localization and interaction with other proteins, including TIMPs [1,6]. MMPs are involved in several mechanisms from cell differentiation, proliferation and angiogenesis to apoptosis, and though they can play a physiological role in many aspects such as skeletal muscle repair and wound healing, they can also be associated with different pathologies, including inflammatory diseases, atherosclerosis, corneal neovascularization, myocardial infarction and cancer progression (to name a few) [1,7,8,9,10,11].

The extracellular activity of MMPs has been well understood, and these enzymes have been subdivided into different groups based on their ECM substrates and/or subcellular locations, including collagenases, gelatinases (A and B), stromelysins, matrilysins, membrane-type (MT), macrophage metalloelastases and epilysins [7,12,13,14,15,16]. However, over the last couple of decades, MMPs have been identified to have intracellular roles as well. The initial discovery of intracellular MMP-2 sparked a surge in research to understand what MMPs have intracellular roles and what those roles are [1,17]. Currently, several MMPs have been found to hold these intracellular roles including MMP-1, MMP-2, MMP-3, MMP-7, MMP-8, MMP-9, MMP-10, MMP-11, MMP-12, MMP-14 (MT1-MMP), MMP-23, and MMP-26 [1,18,19,20]. With these discoveries, we are now able to piece together some of the missing links that make up MMPs.

MMPs are generally excreted via the endoplasmic reticulum as a result of the N-terminal secretory signal [1,21]. However, in HEK293 cells, about half of nascent MMP-2 is retained inside the cell due to an inefficient secretory signal sequence [22]. Two variants of MMP-2 lacking the signal sequence are also found intracellularly: MMP-2_NNT50,_ which lacks the first 50 amino acids [22] and MMP-2_NNT76,_ which lacks the first 76 amino acids [23]. Additionally, MMPs may also undergo endocytosis following secretion, by LDL-related protein receptor binding for MMP-2, MMP-9 and MMP-13 and by caveolae for MT1-MMP [1,24]. Recent research has shown new intracellular locations for MMP-2 and MT1-MMP in various cell types, including cardiomyocytes, megakaryocytes/platelets, retina, immune cells, osteosarcoma, and other cancer cells [1,20,25,26,27].

With increasing research into the intracellular roles of MMPs, there is a rapid influx of new, exciting data that has the potential to not only expand our understanding of this family of enzymes but also impact the future directions of therapy for a variety of disease states. Both MMP-2 and MT1-MMP are ubiquitous in a variety of tissues, and unlike other MMPs, they are constitutively expressed [28]. Figure 1 depicts differences in the structure of MMP domains of MT1-MMP (membrane-type family) and MMP-2 (gelatinase family). In this review, we will summarize the novel subcellular localizations and intracellular roles of MT1-MMP and MMP-2 and explore the future direction of MMP research.

## 2. Subcellular Localization of MT1-MMP

MT1-MMP, a transmembrane protein, is first known to localize to the cell membrane [29]. Recently, the importance of its subcellular localization has been of increased interest [29]. Subcellular mapping of MT1-MMP in the Human Protein Atlas (http://www.proteinatlas.org/ENSG00000157227-MMP14/cell (accessed on June 2022)) revealed that this protease is largely localized to the cytosol and to the intermediate filaments of the cytoskeleton [30]. Markedly, apart from the accumulation of MT1-MMP on the cell surface, MT1-MMP also localizes to the cytoplasm, caveolae, Golgi apparatus, and nucleus [8,13,26,27].

The structure of MT1-MMP, in terms of sequence domains, serves a vital role in directing the protease to extracellular and intracellular compartments, allowing for variability in localization [31] (Figure 1 and Figure 2). The hemopexin-like and cytoplasmic tail domains are involved in trafficking MT1-MMP throughout cellular compartments to the cell membrane, as demonstrated in breast carcinoma MCF7 cells [32,33,34]. Both domains allow MT1-MMP to be internalized to certain regions and organelles within the cell, including the tubulin cytoskeleton and Rab4 positive vesicles in the pericentrosome [32,33,34]. The cytoplasmic tail of MT1-MMP possesses the ability to allocate the protease, through the use of an “up/down” switch, to the cell membrane for the execution of its role in migration and invasion, as shown in HT1080 cells and invasive cancer cells [32,35,36]. Along with the cytoplasmic tail, Urena et al. (1999) identified C-terminal valine as a crucial element of proper trafficking and development of MT1-MMP [37]. Mutations in the C-terminal valine (Val^582^) have resulted in inhibition of specific MT1-MMP processing [37,38]. Evidently, the structure and domains of MT1-MMP play an important role in translocating this protease to multiple intracellular locations [39].

For instance, MT1-MMP is able to internalize inside the cell (i.e., HT1080 fibrosarcoma cells and human endothelial cells) by the use of caveolae, comprised of a small fraction of the plasma membrane formed into a lipid-raft structure, or clathrin-coated pits, through endocytosis [12,32,35,36,40,41,42,43]. Caveolae serve several roles in endocytosis and signal transduction, and they recycle cell surface molecules [32,44]. Hence, caveolae play an important role in translocating MT1-MMP to invadopodia [45]. Studies show that MT1-MMP is initially internalized before relocating to invadopodia through caveolae-mediated endocytosis [45]. This is essential in ensuring a consistent flow of active MT1-MMP to the plasma membrane [46]. Additionally, Rozanov et al. (2004) revealed that when MT1-MMP lacks the C-terminal cytoplasmic tail, this mutant is aberrantly trapped and localized to the caveolae [32]. This localization results in reduced cell migration and tumorigenesis, characterizing the cytoplasmic tail peptide sequence as paramount in the effective release of MT1-MMP from lipid rafts and the transport of the enzyme to the necessary cell surface targets [32,40]. The containment and ineffective release of MT1-MMP from caveolae through the regulation of the cytoplasmic tail indicates one mechanism of regulating this protease [32].

Additionally, endocytosis of MT1-MMP through the use of clathrin and caveolae was recorded inside fibrosarcoma, breast, colon and hepatocellular carcinoma cells [12,47]. When the vital component of caveolae, caveolin-1, was silenced, MT1-MMP’s ability to degrade the extracellular matrix was interrupted in MDA-MB-231 cells [47,48,49]. Consequently, the interaction between the cytoplasmic domain of the protease and Src-mediated tyrosine residue 573 phosphorylation of caveolin-1 further supports the interaction between MT1-MMP and caveolae and the crucial role of caveolae in regulating intracellular trafficking of MT1-MMP and its activity [50].

Furthermore, MT1-MMP localizes to the Golgi apparatus at the perinuclear regions for vesicular transport, as observed in both PC3 and BPH-1 prostate cell lines [51]. An actin and microtubule modulatory protein, LIMK1, is involved in regulating the vesicular trafficking of MT1-MMP for surface localization [51]. Some studies found that LIMK1 serves an essential role in regulating Golgi vesicle transport between the endoplasmic reticulum and Golgi apparatus [51]. When LIMK1 is inhibited, the targeting of MT1-MMP to the plasma membrane is significantly reduced, signifying the role of LIMK1 in MT1-MMP vesicular transport to the cell membrane’s surface [51]. This protease is dependent on signal sequences to direct them to the Golgi or endoplasmic reticulum. Although this is the case, some studies suggest that MT1-MMP lacks efficient signal sequences to effectively direct all of MT1-MMP to cell membrane/secretion [12]. For this reason, splicing, an additional mechanism, may assist in targeting MMPs to subcellular compartments, offering additional regulation [12].

With the further examination of the localization of MT1-MMP inside the cell, MT1-MMP was also discovered to localize inside the nucleus [52]. Ip et al. (2007) examined MT1-MMP subcellular localization in hepatocellular carcinoma, focusing on the nuclear MT1-MMP [52]. When MT1-MMP was found in nuclei from clinical specimens, interestingly, it was correlated with poor survival due to aggressive tumor characteristics [52]. MT1-MMP has a presumed nuclear localization sequence (NLS), translocating the protease to the nucleus [12,53]. However, other MMPs without a NLS were found to translocate to the nucleus via an additional mechanism, e.g., latching onto nuclear-translocating proteins [54,55].

## 3. Novel Roles of MT1-MMP inside Cells

It is relatively known where and how MT1-MMP is localized to subcellular regions throughout the cell [31]. However, revealing the exact functions of this protease in each intracellular region is challenging and important to understanding the entirety of MT1-MMP’s role in several cellular processes [31]. Regulated compartmentalization of MT1-MMP is essential for its wide range of proteolytic-dependent and -independent functions [29]. As reported in several tumors, MT1-MMP has been associated with angiogenesis, cell migration, invasion, and cell growth, resulting in accelerated tumorigenesis [56,57,58,59,60]. Due to these functions, the investigation of intracellular MT1-MMP functions is vital to understanding its role in various pathologies.

## 4. Novel Role in Centrosome Function

The localization of MT1-MMP in the centrosome revealed its ability to cleave the centrosomal protein, pericentrin, an integral component in centrosome function and mitotic spindle formation [33,61]. In mammary epithelial cells, overexpression of MT1-MMP disrupts the centrosome function and results in the formation of abnormal spindles, aneuploidy, chromosome instability and thus, tumorigenesis [33,62]. Another centrosomal protein that is cleaved by MT1-MMP is the breast cancer type 2 susceptibility gene (BRCA2), which indicates a potential role of MT1-MMP in regulating BRCA2 trafficking and genome stability in HeLa, S3, U2OS and MCF7 cells [63]. Thus, the interactions between MT1-MMP and centrosome proteins are important in centrosome function and genome stability.

## 5. Novel Role in Migration

It was discovered that the short intracellular domain of MT1-MMP serves an important role in cell migration and invasion, as it is involved in the intracellular localization to various subcellular compartments, intracellular development of invadopodia and different signal transduction pathways [31,36,64,65,66,67,68]. The cytoplasmic domain of MT1-MMP specifically plays a crucial role in correlating the enzyme’s proteolytic activity to proper cell migration and invasion [31]. The tyrosine residue 573 in MT1-MMP’s cytoplasmic domain is phosphorylated by Src, a critical kinase in cell migration [31]. This is exhibited when cell migration was reduced in cells overexpressing a mutant of MT1-MMP unable to be phosphorylated (Y573F) in human umbilical vein endothelial cells and HT-1080 cells [31,69]. A further inspection of this pathway found the phosphorylation of endophilin A2 by a focal adhesion kinase (FAK) and Src complex leads to less MT1-MMP undergoing endocytosis, resulting in increased migration [50]. Likewise, the phosphorylation of the intracellular domain of MT1-MMP serves an important role in cell migration [31,36]. When MT1-MMP is phosphorylated at Thr567 or Tyr573, it co-localizes with p130Cas, an adaptor protein, at cell protrusions under migratory conditions [69,70,71,72,73,74]. MT1-MMP was also shown to be upregulated with increased growth rates and angiogenesis [31]. Additionally, MT1-MMP is considered a constituent of integrin-triggered adhesion sites, specifically with the interaction of MT1-MMP and phosphorylated caveolin-1 or the adaptor p130Cas, leading to the phosphorylation of MT1-MMP by Src [49,69]. The relationship between MT1-MMP and integrins is important to acknowledge, as integrins are important in modulating growth signaling and cell migration and produce signals that regulate gene transcription [75,76,77,78].

## 6. MT1-MMP Interacts with Tumor Suppressors

MT1-MMP was also found to interact with bona fide or potential tumor suppressors inside the cell. MT1-MMP activity was found to suppress the expression of the tumor suppressor, SPRY4 [79]. Shaverdashvili et al. (2015) inhibited SPRY4 in melanoma cells, which rescued cell migration in MT1-MMP knockdown cells. This data revealed SPRY4’s role in MT1-MMP regulation and thus cell migration [79]. As also observed in metastatic melanoma, high levels of SPRY4 are associated with prolonged survival of melanoma patients [79]. Additionally, the MT1-MMP cytoplasmic-binding protein 1 (MTCBP-1) was found to suppress MT1-MMP mediated cell migration through its interactions with the cytoplasmic domain of MT1-MMP [80,81]. Among various cancer cell lines (U87 glioblastoma cells, brain tumors), data show that MT1-MMP was highly expressed, while MTCBP-1 was minimally expressed, contrasting with normal fibroblasts [40,81]. These results suggest that MTCBP-1 is a potential tumor suppressor that interacts with MT1-MMP inside the cell [40].

## 7. Novel Role of Caveolae in MT1-MMP Recycling

Caveolae, or lipid raft domains, important for endocytosis of MT1-MMP, are not only critical in the internalization of the protease, but they also determine the location and function in different cell types (e.g., HT1080 fibrosarcoma cells) [29,43]. Caveolae are involved in vital cellular processes, including signal transduction, lipid signaling and plasma membrane resistance to mechanical stress, and perhaps of more interest, the role of MT1-MMP in metastasis and migration, especially in several cancer cells [40,71,72,73,82,83]. Cell surface expression of MT1-MMP is low in a few cell types, including MDA-MB-231 cells, due to the rapid endocytosis that results in its intracellular localization, which may be a possible mechanism for caveolae to downregulate MT1-MMP activity [31,35,40,84]. On the other hand, deletion of MT1-MMP cytosolic tail results in its forced allocation to lipid rafts and reduces its access and processing of E-cadherin and tumor cell locomotion and growth [29,32,85]. Thus, these intracellular interactions of MT1-MMP’s cytosolic tail govern various intracellular responses, as it works with caveolae to localize the protease to the necessary subcellular locations [29,31].

## 8. Novel Role of MT1-MMP in Invadopodia and Podosome Formation

Endocytic and exocytic pathways may be involved in the polarized delivery of MT1-MMP to invadopodia, leading to its rapid formation [44]. Invadopodia initially begin their formation with the aggregation of actin and cortactin at areas of cell adherence to the extracellular matrix [86,87]. MT1-MMP also begins to accumulate at these initiation sites, and as invadopodia mature, MT1-MMP, actin and cortactin levels are increased at these sites [44,50]. Intact microtubules and the intermediate filament, vimentin, are necessary for the further elongation of mature invadopodia [88]. Vimentin interacts with the cytoplasmic tail of MT1-MMP to promote MT1-MMP translocation to the endothelial cell surface for ECM degradation [89]. Further, the Golgi complex was found to be located close to invadopodia within the cell [44,90]. This location allows for *β*1 integrin adhesion to collagen I fibrils to trigger rapid Rab8-dependent polarized exocytosis to efficiently maintain active extracellular matrix degradation at invadopodia sites, as observed in MDA-MB-231 cells [44,76]. Although invadopodia serve crucial roles in migration, fluorescence recovery after photobleaching (FRAP) analysis revealed MT1-MMP localized to invadopodia possessed limited mobility, while the non-invadopodia plasma membrane regions have increased MT1-MMP mobility and high internalization [17]. Consequently, this indicates a complex dynamic role of MT1-MMP in the development of invadopodia and cell migration.

Accordingly, it is important to mention the role of MT1-MMP in podosome development. Podosomes are actin-rich cell adhesions consisting of Arp2/3 complex-nucleated F-actin and actin-associated proteins and a ring structure composed of the proteins vinculin, talin and paxillin [91,92]. Podosomes interact and communicate with invadopodia to promote cell migration and invasion through degradation of the extracellular matrix [93]. Azzouzi et al. (2016) highlighted the significance of not only the accumulation of MT1-MMP at podosome sites but at these “islets” within the plasma membrane of macrophages [93]. The ability of the C-terminal cytoplasmic tail of MT1-MMP to bind to the cortical actin cytoskeleton contributes to podosome redeveloping at these sites [93]. Moreover, the islets of MT1-MMP play a major role in podosome reemergence and regulation of podosome spacing or dispersion [93].

## 9. Non-Proteolytic Role in Macrophage Regulation and Metabolism

MT1-MMP’s localization to the Golgi apparatus is important for macrophage metabolism, as the cytosolic tail is essential for metabolism regulation [29]. It was shown that MT1-MMP, independent of its protease activity, is able to regulate hypoxia-inducible factor 1 (HIF-1) metabolism, especially in a normoxic environment [94]. The cytoplasmic domain of MT1-MMP binds to factor inhibiting HIF-1 (FIH-1), in turn allowing Mint3/APBA3 to inhibit FIH-1 [94]. Likewise, the MT1-MMP/FIH-1/Mint-3 complex is formed at the Golgi, resulting in the release of HIF-1α and activating glycolysis for ATP production in human fibrosarcoma cells (HT1080) [4]. The interaction of MT1-MMP may coordinate its invasive properties with HIF-1α, as the cellular metabolism is also critical for macrophage cell migration [29]. Specifically, in tumor cells, the peripheral Golgi matrix protein, GRASP55, interacts with the LLY^573^ motif in the cytoplasmic domain to regulate the exocytosis of MT1-MMP [60]. The importance of this motif is emphasized when the LLY^573^ motif is mutated, as the mutation significantly reduces the internalization of MT1-MMP [60]. Therefore, the presence of MT1-MMP at the Golgi apparatus is essential for metabolism reprograming, which impacts macrophage motility and migration.

## 10. Novel Role in Gene Regulation

Of particular interest is the function of MT1-MMP within the nucleus. When MT1-MMP expression is upregulated or silenced, the transcription of several genes is differentially expressed [30,31]. This is observed in the down-regulation of Smad1 upon knockdown of MT1-MMP, resulting in reduced tumor growth, as well as the upregulation of the tumor suppressor Dickkopf-3 in human urothelial carcinoma cells and VEGF-A in MCF-7 and U251 cells [95,96,97]. These results suggest that MT1-MMP may be involved in regulating the transcription of several genes involved in migration and metastasis. Nuclear translocation of MT1-MMP was speculated to regulate chromatin remodeling directly or indirectly and thus gene transcription in endothelial cells [29]. In cancer cells, due to their tendency to exhibit chromosomal instability leading to malignancy, it is suggested that MT1-MMP’s localization to the centromere plays a role in the development of mitotic spindle abnormalities and chromosome instability, leading to increased invasion and migration [33,62].

Additionally, genome-wide expression profiling was completed on cancer cells overexpressing or silencing MT1-MMP. These studies revealed several genes that are strongly associated with MT1-MMP levels, including regulators of energy metabolism, signaling and transcription, chromatin rearrangement, cell division, and apoptosis [31,68]. A ChIP and reporter vector analysis showed that MT1-MMP regulates the expression of more than 100 genes [12]. Of those genes, 20% are connected to immune regulation, independent of MT1-MMP’s proteolytic functions, also strongly inferring a role in macrophage inflammatory responses [26]. The transcription of phosphoinositide 3-kinase δ (PI3Kδ) is correlated to the translocation of MT1-MMP to the nucleus [26]. Once translocated to the nucleus, nuclear MT1-MMP binds to the p110 promoter to activate the transcription of PI3Kδ, resulting in an Akt/GSK3 signaling cascade that forms the Mi-2/NuRD complex [26]. Among MT1-MMP null macrophages, Shimizu-Hirota et al. (2012) revealed defects in PI3Kδ/Akt/GSK3 activity, Mi-2/NuRD expression, and pro-inflammatory gene regulation. This was restored when a catalytically inactive form of MT1-MMP was re-expressed, suggesting MT1-MMP plays an important non-proteolytic role in macrophage gene regulation [26]. Due to the indirect interaction between MT1-MMP and the ATP-dependent nucleosome remodeling complex, Mi-2/NuRD, more insights into the transcription regulations and epigenetic functions of nuclear MT1-MMP could be further revealed [26,29]. Future research will identify signaling pathways correlated with MT1-MMP’s intracellular locations and its potential nuclear function in transcription regulation, which will unveil critical mechanisms involved in tumorigenicity and metastasis of various cancer cells. Figure 2 summarizes novel intracellular locations/roles of MT1-MMP in various cellular processes.

## 11. Subcellular Localizations of MMP-2

MT1-MMP was not the first MMP to be found within intracellular compartments of the cell [1]. In fact, MMP-2 is also regulated, in part, by intracellular vesicles, or caveolae, as they are translocated to different subcellular locations [98,99]. MMP-2 localizes to specific intracellular compartments, including the cytosol, mitochondria, and nucleus, as first discovered to function inside cardiomyocytes and later inside other cell types, including cancer cells [12,53,100,101].

Of the intracellular compartments, MMP-2 possesses a tendency to remain in the cytosol since half of nascent MMP-2 is retained inside the cell [22]. The inefficiency of MMP-2’s secretory signal may be attributed to the evolutionarily conserved feature of MMP-2 homologues, as demonstrated in zebrafish [102]. Fallata et al. (2019) investigated the localization of MMP-2 to the M-band of skeletal muscle sarcomeres and revealed the potential existence of selective pressure against the efficient secretion of MMP-2 [102]. Conversely, despite the presence of a signal sequence that translocates MMP-2 into the endoplasmic reticulum (ER) for secretion, a significant portion of the protease is retained in the cytosol [22]. We previously studied the effect of an increase in cytosolic MMP-2 in neonatal cardiomyocytes exposed to oxidative stress [22]. When a transcript variant of MMP-2 that lacks the first 50 amino acids (MMP-2_NTT50_) was increased, a cardiac sarcomere protein, Tnl, experienced increased degradation in rat hearts [22]. These results demonstrate the importance of regulating both canonical and other MMP-2 variants in health and disease [22]. Active research in our laboratory is ongoing in order to investigate MMP-2 domains responsible for the transport of MMP-2 to various subcellular locations. This research may provide insight into the mechanisms responsible for regulating levels of MMP-2 secreted and those translocated to other subcellular compartments, including the nucleus.

Although a significant portion of MMP-2 resides in the cytosol, MMP-2 is the first MMP to be detected within the nucleus [12,103]. Several studies detected MMP-2 inside nuclei of a wide range of cells: rat liver, human heart, hepatocellular carcinoma, pulmonary artery endothelial cells undergoing apoptosis, and brain cells [12,52,103,104,105,106]. Additionally, some studies have identified MMP-2 as the only MMP within the nuclei of epithelial cells and primary neurons [107]. The reoccurring pattern of nuclear MMP-2 across various sample types highly suggests that MMP-2 possesses a crucial role in gene regulation [12]. In addition to determining the exact role of nuclear MMP-2, it is also necessary to understand how nuclear MMP-2 is regulated.

As observed with MT1-MMP, MMP-2 contains a nuclear localization sequence (NLS) to regulate intracellular trafficking into the nucleus and may even specify subnuclear localization, e.g., targeting nucleoli [107,108]. Transport of MMP-2 to the nucleus occurs through the process of receptor-mediated nuclear shuttling due to the presence of an NLS [108,109]. The NLS allows importins α and *β* to recognize and bind to the sequence, forming an importin-cargo complex [110]. The complex is then able to bind to the nuclear pore complex, mediating the translocation of proteins from the cytoplasm into the nucleus [108,111]. There are at least two types of NLS: the classical NLS and the proline-tyrosine NLS [108,112,113]. Each possesses sequences corresponding to specific importin proteins [108]. In the case of MMP-2, in silico analyses revealed classical NLS, as it is responsible for the nuclear translocation of MMP-2 [103,108]. Currently, the NLS of MMP-2 has not been experimentally identified. However, the research conducted on the NLS for MMP-3 revealed the existence of similar NLS along with 5 putative NLS throughout the MMP-3 domains [108,114]. When each one of these NLS was expressed, MMP-3 was transported to the nucleus in chondrocytes [53,108,114]. As a result of these findings, multiple NLS infer the potential for post-translational modified MMPs to cloak the primary NLS and expose one of the putative NLS, suggesting mechanisms that offer increased selectivity of nuclear shuttling [53,108,114]. Current research by our group is ongoing to determine whether MMP-2 possesses multiple NLS and to uncover other potential mechanisms responsible for translocating MMP-2 to the nucleus and/or nucleolus.

Furthermore, post-translational modifications of MMP-2 may affect its nuclear localization [61]. In general, the transcription of MMP genes is known to be regulated by epigenetic mechanisms, such as DNA methylation and histone acetylation [115]. The regulation of MMP expression post-transcriptionally is mediated by regulating mRNA stability and miRNA-based mechanisms that regulate MMP expression [115,116,117]. The post-translation level of MMP is currently considered one of the most important levels of regulation as it is the reason for the secretion of the majority of MMPs as inactive proenzymes [115,118]. Pro-MMP activation may be prompted by their interaction with extracellular matrix proteins and cell surface molecules [115]. Sariahmetoglu et al. (2007) specifically demonstrated that phosphorylation of MMP-2 by protein kinase C (PKC) significantly impacts the activity of this protease [119]. Furthermore, MMP-2 may also undergo post-translational modifications in the Golgi apparatus, located in close proximity to the nucleus [61]. This is demonstrated by furin possessing the ability to cleave pro-MMP-2 within the trans-Golgi network, albeit inactivating the protease [120]. There is evidence that MMP-2 is purposefully translocated to the nucleus, and its translocation does not occur through accidental invasion of the Golgi to nuclear pores [61]. Evidently, MMP-2 contains several mechanisms to transport inside the nucleus, suggesting that nuclear MMP-2 plays a key role in regulating nuclear matrix remodeling and gene expression [12].

## 12. Novel Roles of MMP-2 inside Cells

With research providing more insight into the potential roles of intracellular MMP-2, a growing number of crucial functions have been revealed and parallel with, in a broad sense, intracellular roles of MT1-MMP [103]. These functions include cell migration, proliferation, calcium handling, muscle contraction, gene expression and ribosomal RNA transcription [103,108] (Figure 3).

## 13. Novel Role in Cardiac Dysfunction

Under certain conditions, MMP-2 can function as a result of intracellular activation or the result of being free of tissue inhibitors of metalloproteases (TIMPs) [121]. TIMPs are endogenous inhibitors that regulate MMPs once the proteases are activated. [115,122]. Inside cardiomyocytes, MMP-2 was found to co-localize to the sarcomere and cytoskeleton [123]. MMP-2 is activated upon oxidative stress and proteolyzes various sarcomeric proteins, including TnI, α-actinin and titin, which contributes to cardiac contractile dysfunction [123,124]. MMP-2 also cleaves and activates GSK-3β in the cytosol, which may contribute to cardiac damage [125]. Consequently, inhibiting intracellular MMP-2 activities in hearts during ischemia/reperfusion was found to exert cardioprotective effects [126,127]. It is worth mentioning that TIMP-4 was also found within thin myofilaments inside cardiomyocytes, which indicates a potential regulatory role of TIMPs inside cells [128].

## 14. Novel Roles of MMP-2 inside the Nucleus

MMP-2 was the first MMP to be detected inside the nucleus of cardiac myocytes [103]. Several of the functions and mechanisms of nuclear MMP-2 remain unclear, albeit there is evidence suggesting the potential involvement in nuclear matrix remodeling and degradation of nuclear proteins, including transcription factors and RNA processing proteins [61,107]. Nuclear MMP-2 was revealed to be an active form that has undergone cleaving of the proenzyme [61]. A study compared the NLS of a wide range of MMPs and revealed some MMPs possessed a conserved NLS [113]. It is possible that nuclear MMP-2 may function similarly to extracellular MMP-2 or may have evolved independently within the nucleus [129]. Likewise, TIMP-1 was found to translocate to the nucleus of human MCF-7 breast carcinoma cells after binding to the cell surface [103]. This response of TIMP-1 immediately translocating to the nucleus implies an important role of MMP-2 within the nucleus, requiring regulation, especially within cancer cells [103].

In parallel with the role of MT1-MMP within the nucleus, it is suggested that MMP-2 possesses the ability to regulate certain genes and is involved in their transcription through proteolytic activity [103,108]. Proteolytic cleavage of the nuclear matrix is important in the following processes: apoptosis, regulation of the cell cycle and nuclear matrix degradation [103]. Nuclear matrix degradation is also fundamental in DNA fragmentation and condensation of chromatin [103]. Interestingly, MMP-2 co-localizes with poly-ADP ribose polymerase-1 (PARP-1) within the nuclear matrix, a structure similar to the extracellular matrix as it provides structural integrity and support for vital cellular processes [61,103,130]. PARP-1 is activated by single-strand DNA breaks and serves a vital role in the DNA damage response pathway to repair the single-strand breaks [61,130]. There is evidence that nuclear MMP-2 plays a role in cleaving PARP-1 in cardiac myocytes, which can be inactivated by proteolytic cleavage [12,103]. Further supporting MMP-2’s role in cleaving PARP-1, another study showed that cigarette smoke increases the expression and nuclear localization of MMP-2 in apoptotic endothelial cells, which exhibited proteolytic activity within the nucleus cleaving PARP-1 [104]. Inhibiting DNA repair through the cleavage of PARP-1 by MMP-2 results in apoptosis [131]. On the other hand, excessive PARP-1 activation may induce energy depletion of cells, leading to apoptosis [61].

Although MMP-2 is localized to other intracellular compartments, data suggest that nuclear MMP-2 is crucial to carrying out cellular processes. A study was conducted to examine the effect of treating endothelial cells with Golgi endosomal structure-disrupting agents, which inhibit protein synthesis and development of microtubules, on MMP-2 levels [107]. Immunofluorescence, Western blotting and immunoprecipitation of nuclear MMP-2 revealed the lack of impact of disrupting the Golgi structure on the pattern and level of MMP-2 within the nucleus, thus highlighting the stability of nuclear MMP-2 [61,107]. As a result of these findings, studies suggest that MMP-2 localizes to the nucleus through other factors and interactions other than processing at the Golgi apparatus [61,107].

## 15. Novel Role in Ribosomal RNA Transcription

We recently examined MMP-2 in a designated subnuclear structure, the nucleolus, in osteosarcoma cells [22]. The nucleolus serves a role in housing the ribosomal RNA for transcription and in the biogenesis of ribosomes [22]. The localization of MMP-2 to the promotor region of the ribosomal DNA genes in the nucleolus suggests it regulates ribosomal RNA transcription [22]. Indeed inhibition of MMP-2 genetically or pharmacologically was associated with reduced both pre-ribosomal RNA transcription and cell proliferation [27]. We also found that nucleolar MMP-2 activity regulates RNA transcription through the clipping of histone H3. Inside the nucleolus, histone H3 cleavage is associated with increased ribosomal RNA transcription. Under MMP-2 inhibition, there is minimal, or a lack of, histone H3 cleavage, and this results in a reduction in transcription, as well as cell proliferation [27]. Another study also found that MMP-9 (Gelatinase-B, closely related to MMP-2) is able to clip histone H3 and regulate gene expression involved in melanoma formation [132]. These studies further support MMP-2’s involvement in the epigenetic regulation of gene transcription within not only the nucleolus, but moreover, the nucleus.

## 16. Potential Interaction of MT1-MMP and MMP-2 inside the Cell

MT1-MMP is closely related to MMP-2 activation. Both are constitutively expressed in various cell types and overexpressed in most cancers [28]. As a result of these findings, the subcellular and nuclear localization of MT1-MMP was clinically examined in hepatocellular carcinoma specimens [52]. As both MT1-MMP and MMP-2 exist inside the nucleus, we speculate that MT1-MMP may also be responsible for the activation of MMP-2 in the nucleus, leading to the activation of several nuclear functions, similar to the previous mechanism of activation described outside the cell [133]. Further, as pro-MMP-2 is activated by MT1-MMP, it is possible that active MMP-2 cleaves MT1-MMP and regulates its activity and subcellular localization [134,135]. The presence of both MT1-MMP and MMP-2 within the nucleus highlights the importance of these novel roles and interactions between both MMPs inside the cell.

Furthermore, one target within the nuclear matrix includes fibronectin, which is a substrate of both MMP-2 and MT1-MMP. It is a very important factor in cancer metastasis. These observations suggest potential interactions for both MMPs within the nucleus [136,137]. It is quite tempting to speculate that MT1-MMP and MMP-2 may interact and regulate the gene expression of each other inside the nucleus. Future studies need to examine the effect of nuclear MT1-MMP or nuclear MMP-2 on the expression of other MMPs genes as well. Examining the roles of MT1-MMP and MMP-2 within the nucleus and how they potentially interact may provide new insights into the roles of nuclear MMPs critical in regulating cancer epigenetics and tumor migration and invasion [52].

One potential method to target MMP-2 and MT1-MMP within the nucleus includes the use of cell-penetrating peptides (CPPs), as they are able to penetrate the cell membrane without the need for specific receptors or transporters [138]. A study completed by Hariton-Gazal et al. (2002) demonstrated the use of a peptide derived from the dermaseptin family with cell-penetrating properties [138]. A peptide derived from dermaspetin was directed to the nucleus of HeLa cells by covalently attaching peptides containing NLS [138]. Zhang et al. (2016) also used CPP, CB5005, with the addition of an NLS (CB5005N) to target tumor spheroids of U87 cells and inhibit tumor growth. When CB5005N was used in combination with doxorubicin, there was a synergistic anti-tumor effect [139]. Moreover, a study highlighted the use of antibodies modified with an NLS peptide to determine the efficiency of nuclear targeting activity of 11In-trastuzumab-NLS in human gastric cancer cells [140]. Keilko Li and Hasegawa (2022) suggested the use of this system may be able to effectively target nuclear HER2 [140]. Accordingly, modifying anti-MMP-2 and anti-MT1-MMP antibodies to contain NLS will allow us to target nuclear MMPs. As a component of our future research, we aim to fully develop this method for nuclear MMP-2.

## 17. Conclusions and Future Directions

Despite its first discovery as membrane-bound or secreted proteases, mounting evidence shows that MT1-MMP and MMP-2 are present and active in various subcellular locales. Future research is strongly needed to identify new intracellular substrates and epigenetic functions of MT1-MMP and MMP-2 in different diseases. For instance, nuclear MT1-MMP or nuclear/nucleolar MMP-2 may be defined as novel therapeutic targets for metastatic cancer. This is important because old-generation MMP inhibitors have been designed and tested based only on the extracellular role of MMPs. These inhibitors were non-selective and failed clinical trials in cancer therapy mainly due to dose-limiting toxicities. Therefore, unravelling novel roles of MT1-MMP and MMP-2 beyond the extracellular matrix will be significant because they are expected to offer new mechanistic insights into different pathologies. We recently discovered that nuclear/nucleolar MMP-2 regulates ribosomal DNA genes and osteosarcoma migration [27]. Understanding how targeting nuclear MT1-MMP or MMP-2 affects cell proliferation and migration will open new avenues in cancer therapy. This will help better design more specific inhibitors targeting subcellular MT1-MMP/MMP-2 to improve the efficacy of therapeutic interventions in cancer and other diseases.

## Figures and Tables

**Figure 1 ijms-23-09513-f001:**
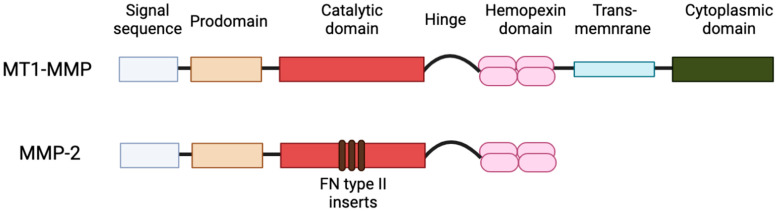
Structure of MMP domains of a membrane-type (MT1-MMP) and a gelatinase (MMP-2) member of MMPs. MT1-MMP has transmembrane and cytoplasmic domains that are absent in MMP-2. MMP-2 has fibronectin (FN) type-II inserts within its catalytic domain.

**Figure 2 ijms-23-09513-f002:**
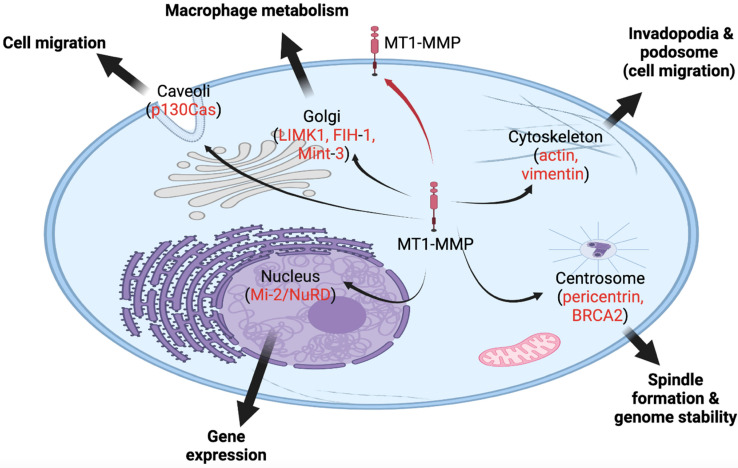
MT1-MMP, a membrane-bound protease, is also found in various intracellular locales, and it functions in various cellular processes inside the cell. The red arrow indicates the canonical trafficking of MT1-MMP to the cell membrane. Black arrows indicate novel intracellular locales of MT1-MMP to caveola, Golgi, cytoskeleton, centrosome, and nucleus. In red, potential substrates or partners of MT1-MMP are listed in these locations.

**Figure 3 ijms-23-09513-f003:**
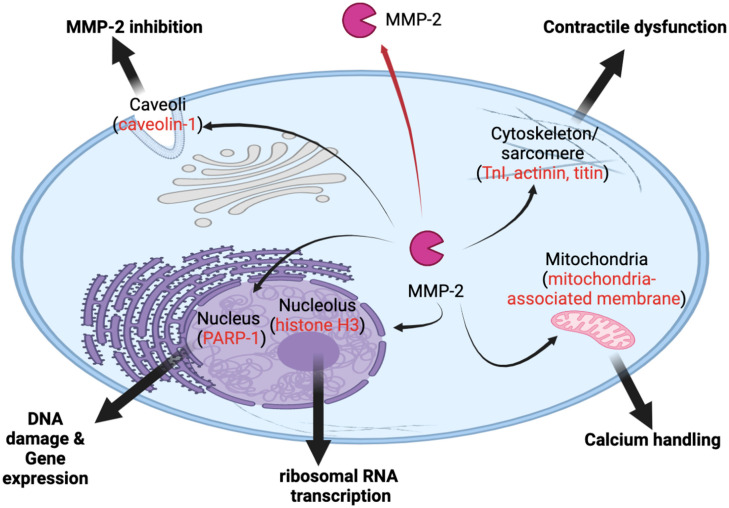
MMP-2 was first described as a secreted protease; however, MMP-2 is localized to prominent subcellular locales and functions at these locations. The red arrow indicates the canonical trafficking of MMP-2 outside the cell. Black arrows indicate new intracellular locales of MMP-2 to caveola, sarcomere, cytoskeleton, mitochondria, nucleolus, and nucleus. In red, potential substrates or partners of MMP-2 are listed in these locations.

## Data Availability

Not applicable.

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
