# Peer review of "Novel Roles of MT1-MMP and MMP-2: Beyond the Extracellular Milieu"

_ijms, 2022, doi:10.3390/ijms23179513_

Round 1
Reviewer 1 Report
This mini-review is promising and strategic in covering non-ECM roles of MT1-MMP and MMP-2, i.e., intracellular localization and trafficking. For example, the summary of MT1-MMP recycling is the greatest strength and contribution of this manuscript. I found the summary of connections to energy metabolism, the centrosome, and PARP-1 interesting as well. Some other aspects of the mini-review in its current form are disappointing and make me view it as not ready for publication, however:
1) I was intrigued by the closing statements suggesting targeting nuclear MT1-MMP and MMP-2 in cancer cell proliferation and migration. Insufficient development of this idea made the idea seem hollow and disappointing upon reading the manuscript.
a) There is so much literature on the ECM activities of these proteases in cancer cell proliferation and migration, that the authors would need to make a stronger case for the importance of the nuclear activities, or offer a strategy for testing that.
b) How could specifically nuclear MT1-MMP and MMP-2 be targeted anyway? Ideas and strategies for doing so would help the field be more open to the testing of this suggestion.
c) Without such improvements, I’d recommend dropping the claim about nuclear targeting.
2) The non-proteolytic activities aren’t developed adequately. Ref 5 does a much better job of this. Ref 66 is one of the papers replete with such activities.
3) There aren’t enough molecular specifics. For example, not much of the rich content of ref 66 is mentioned. Figures 1 and 2 don’t mention strategic molecular partners outlined in the text. Instead, Figs. 1 and 2 look like textbook figure about cell compartments with arrows added. The current forms of Figs. 1 and 2 represent missed opportunities.
4) The original research papers are cited too little and the reviews are cited too much. For example, the text at the bottom of p. 5 seems to attribute the characterization of the interaction with Mi-2/NuRD complexes to ref 5 (a review) instead of the original report in ref 66. (Did the authors read ref 66?) I (and some other journals) urge shifting to citing the original articles more and the reviews less.
5) There is some repetition in the first 4 pages. VEGF passages are repeated, for example.
6) There are too many instances of clumsy phrasing and incorrect grammar. An incomplete list includes:
p.6: “MMP-2 do not “
p. 6: “rat hears”
p. 6, 8: “MMP-2 is the first MMP to be “
p. 6: ‘genes regulation”
p. 8: “or evolved”
p. 8: Run-on sentence too long for me to decipher: “We also found that nucleolar MMP-2 activity regulates RNA transcription through the clip-ping of histone H3, as H3 cleavage is associated with increased ribosomal RNA transcrip-tion and minimal or absence of H3 cleavage, when MMP-2 is inhibited, results in reduc-tion in transcription, as well as cell proliferation [87].”
p. 10: “Despite first discovered as memebrane-bound”
p. 10: “MT1-MMP and MMP-2 are exsist”
Author Response
"Please see the attachment."

Reviewer 2 Report
This topical review is focused on the intracellular roles of the matrix metalloproteinases MT1-MMP and MMP-2. In particular, the authors highlight their recent work on the presence of MMP-2 in the nucleus and its implications for cellular functions such as gene transcription and signalling.
The authors have chosen an interesting and not often discussed topic, namely intracellular activity of MMPs, and have written a comprehensive article citing a variety of relevant papers. Still, the review would benefit from including several other articles, a brush-up of the figures, and especially a more stringent presentation and discussion of the literature, as outlined below.
Major points
1) The authors repeatedly discuss the role of individual domains of both MMPs. It thus seems to be essential to include a diagram showing the different domains of MT1-MMP and MMP-2.
2) For every result cited, please add the specific cell type that has been used to demonstrate this. This is already done for many results, but not for all (e.g. p. 2: “half of nascent MMP-2 is retained insdie the cell…”, and other instances).
3) The authors mention use of the Human Protein Atlas, where MT1-MMP is being associated with intermediate filaments. However, this is not followed up, and only actin and microtubules are then mentioned as relevant cytoskeletal elements.
4) Trafficking of MT1-MMP to invadopodia, together with respective RabGTPases is discussed. However, there is also literature about MT1-MMP in podosomes, together with respective regulators, that should be mentioned.
5) Specific amino acid residues in the MT1-MMP C-terminus (valine, tyrosine) are mentioned. Please include the actual amino acid residue number.
4) p.4: Interaction of MT1-MMP with tumor suppressors is discussed. The paragraph then starts with MTCBP-1. The reader would thus expect that this is a bone fide tumor suppressor. However, it is only mentioned later that it is only a potential tumor suppressor. Please rearrange this paragraph.
5) “Novel roles of MT1-MMP inside cells”: This paragraph provides a lot of material that is not always presented in a stringent way. For better readability, it should be more structured, and introducing additional subheadings should be one way to do this.
6) Figures: From the figures, it looks like MT1-MMP originates at the plasma membrane and MMP-2 does so from the extracellular space, and that both are only later trafficking to the ER and other organelles. This should be corrected and the real flow of the proteases should be shown.
Also, an arrow points from caveolae to cell migration, while a completely independent one points from the cytoskeleton to invadopodia. This suggests that the cytoskeleton has no connection to cell migration, which it obviously does.
In addition, also add podosomes, as discussed in point 4.
It is unclear what the differently sized and coloured dots represent. Ribosomes and vesicles? If so, please clearly indicate this, as well as their respective roles in MMP trafficking.
7) p.9: The authors speculate that “MT1-MMP may also be responsible for activation of MMP-2 in the nucleus” and close the paragraph by introducing “these novel roles” of the MMP. Please clarify whether the role is speculative or actually demonstrated.
Minor points:
1) p. 2, 3rd paragraph: “highly ubiquitous in most tissues”. This makes no sense, as a protein is either ubiquitous or not.
2) p.4: Dickkopf NOT Dickkof
3) p. 6: “MMP-2 does not translocate” NOT “MMP-2 do not translocate”
4) p.6: rat hearts NOT rat hears
5) p. 7: “When each NLS was expressed”: Please clarify
6) conclusions: “are exsist”
Author Response
"Please see the attachment."

Round 2
Reviewer 1 Report
The revision by Maybee, Ink, and Ali has addressed the main concerns. The claim of targeting the intracellular enzymes has been substantiated with a strategy. (I don’t require the authors to reveal their future plans. I recommend being judicious about this.) Many more original papers are cited appropriately. The figures are more worthwhile now.
Some lingering minor details remain to be corrected:
p. 15, lines 19 and 24: Rather than citing the review (ref 30), the original paper and its expert authors should be cited.
p. 3, line 11: Correct the figure number.
English problems remain to be fixed:
Legends to both Figures 2 and 3: The last “sentence” of each lacks a verb to make it a sentence. Add the verb.
Correct these phrasings, and get someone to read the fixes for correctness:
P. 1, lines 14, 15: two MMPs closely related
p. 2, line 5: last couple decades
p. 2, line 25: ubiquitous in most tissues
P. 2, lines 26-27, 31: structural domain
p. 10, line 29: withing
p. 15, line 8: fund
It looks like the PI needs someone to proofread his writing.
Reviewer 2 Report
The authors have responded to all my concerns.
There are two new typos/misnamers I found:
p. 5, line 51: "Podpsome"
P6, linde 28: "disbursement" should probably read "dispersement"
